# Linking photoelectron circular dichroism to the asymmetric total photoemission yield measured in aerosol nanoparticles of tyrosine

Sebastian Hartweg [1,2,4] ✉, Dušan K. Božanić [3,4], Gustavo A. Garcia [2] & Laurent Nahon [2]

Spectroscopic techniques that are sensitive to molecular chirality are important analytical tools to quantitatively determine enantiomeric excess and purity of chiral molecular samples. Many chiroptical processes however produce weak enantio-specific asymmetries due to their origin relying on weak magnetic dipole or electric quadrupole effects. Photoelectron circular dichroism (PECD) in contrast, is an intense effect, that is fully contained in the electric dipole description of light matter interaction and creates a chiral asymmetry in the photoelectron angular distribution. Here, we demonstrate that this chiral signature in the angular distribution of emitted electrons can be translated into the total photoemission yield for particulate matter. The resulting chiral asymmetry of the photoemission yield (CAPY), mediated by the attenuation of light within condensed particles, can be detected experimentally without requiring high vacuum systems and electron spectrometers. This effect can be exploited as an analytical tool with high sensitivity to chirality and enantiopurity for studies of chiral organic and hybrid submicron particles in environmental, biomedical or catalytic applications.

Photoelectron circular dichroism is a chiroptical effect that creates a forward-backward asymmetry in the photoemission from chiral molecules irradiated with circularly polarized light (CPL)[1-4]. The asymmetry observed in the angular distribution of the photoelectrons depends on the orbital from which the electron is emitted, and on the scattering of the electron off the chiral potential of the molecular cation. As such, PECD is highly sensitive not just to the handedness of a given molecule but also to the molecular conformation and environment[5-10]. The laboratory frame photoelectron angular distribution created by single-photon ionization is written as

$$I(\theta) = b_0(1 + b_1^p \cos\theta + b_2^p(3\cos^2\theta - 1)) \qquad (1)$$

where the kinetic energy dependences of $b_i$ have been removed for clarity, and the polar angle $\theta$ is measured between the propagation axis of the circularly polarized light and the emitted electrons. The first term $b_0 = \frac{\sigma}{4\pi}$ is the isotropic photoionization cross section, describing the photoelectron spectrum (PES). The parameter $b_2^p$ describes the achiral anisotropy parameter, and the dichroic parameter $b_1^p$ quantifies the PECD effect. The superscript $p = 0, +1, -1$ indicates linear, left- and right-handed circular polarization. The dichroic parameter $b_1^{+1} = -b_1^{-1}$ changes its sign when the handedness of either the light polarization or the enantiomer is swapped, while the parameter $b_2^{\pm 1} = -\frac{1}{2}b_2^0$ is not affected by these changes. PECD can be considered universal with respect to the photon energy range and has been observed for single photon ionization from the valence shell using VUV

[1]Institute of Physics, University of Freiburg, Freiburg, Germany. [2]Synchrotron Soleil, l'Orme des Merisiers, St. Aubin, France. [3]Center of Excellence for Photoconversion, Vinča Institute of Nuclear Sciences - National Institute of the Republic of Serbia, University of Belgrade, Belgrade, Serbia. [4]These authors contributed equally: Sebastian Hartweg, Dušan K. Božanić. ✉e-mail: sebastian.hartweg@physik.uni-freiburg.de

and XUV radiation[11] as well as for core shells using X-rays[12,13]. Also, multiphoton ionization PECD experiments[14], including above threshold ionization[15], using visible and UV photons have been reported. Recent developments in the production of pulsed sources of circularly polarized light have put time-resolved PECD studies of electronic and nuclear dynamics[16–21] in the spotlight.

In contrast to other circular dichroism effects, PECD is fully contained in the electric dipole (E1) approximation and not relying on weaker quadrupole (E1E2) or magnetic dipole interactions (E1M1). Therefore, PECD typically exceeds other circular dichroism effects in magnitude, with asymmetries surpassing 10% for various chiral molecules such as terpenes[22–24], oxirane derivates[25,26], organometallic complexes[27] and amino-acids[28]. These large asymmetries make PECD very appealing for analytical purposes, for example, in the pharmaceutical, food and fragrance industries, where the sensitive distinction between enantiomers is a challenge of utmost importance. Analytical applications of PECD are to some extent inhibited by the rather complex experimental setups, including high vacuum systems and electron spectroscopy often associated with imaging techniques, necessary for its detection. Also, except for anionic samples[29,30], and for a few studies on neutral amino-acids[6,28,31,32], most experimental measurements so far have been performed on chiral model systems easy to bring in the gas phase, unlike pharmaceutically- and biologically-relevant substances that are challenging to vaporize due to their limited thermal stability. Finally, note that it is only recently that PECD studies have been reported for liquid microjets[33,34] and condensed aerosol particles in the 100 nm size range[6]. The latter two achievements create the possibility to study PECD in the solution phase, mimicking in vivo conditions, as well as in solid environments.

The large spatial extent of these condensed samples also creates additional achiral anisotropies due to the inhomogeneous distribution of light intensity in the sample volume[35–37]. Typically, the absorption of light in the condensed environments attenuates the light, leading to higher light intensities on the side of the sample that is directly exposed to the radiation, thus producing a higher probability for ionization in these regions. Since isotropic photoelectrons can only escape from a thin surface layer of a few nanometers within the sample, the light intensity distribution is mapped onto the angular distribution of photoelectrons, see schematic representation in Fig. 1a for the case of nanoparticles (NPs) on which we will focus from now on. Photoelectrons created with an initial momentum toward the inside of the particle will be reabsorbed within the particle, and only electrons created with an initial momentum toward the surface will be able to escape from the particle. The consequence is a shadowing effect in the photoelectron angular distribution, with fewer photoelectrons emitted in the forward direction than in the backward direction. Shadowing is often quantitatively described by the shadowing parameter $\alpha = \frac{I_{\text{forward}}}{I_{\text{backward}}}$[35], defined as the ratio of the electron intensities in the forward and backward directions. In Fig. 1a this asymmetry is indicated by the number of black arrows, indicating electrons emitted in a certain direction. The shadowing effect is typically not affected significantly by light polarization or the chirality of the sample. This means the weak chiral asymmetry in absorption cross sections caused by circular dichroism (CD) effects on the order of 0.001%-0.1%[38,39], has only vanishingly small effects on the shadowing parameter. Therefore, the PECD of chiral nanoparticles can still be obtained from differences of photoelectron images obtained with left and right-handed circularly polarized light[6].

Isolated nanoparticles were also the subject of two pioneering chiroptical studies[40,41], before the first experimental discovery[3] or quantitative prediction of PECD effects[2]. In these studies, Paul and Siegmann et al. reported large circular dichroism effects in the total photoemission cross section of nanoparticles of chiral molecules exposed to far UV radiation (193 nm, i.e., 6.42 eV). In the case of tyrosine nanoparticles, the observed chiral asymmetry of the

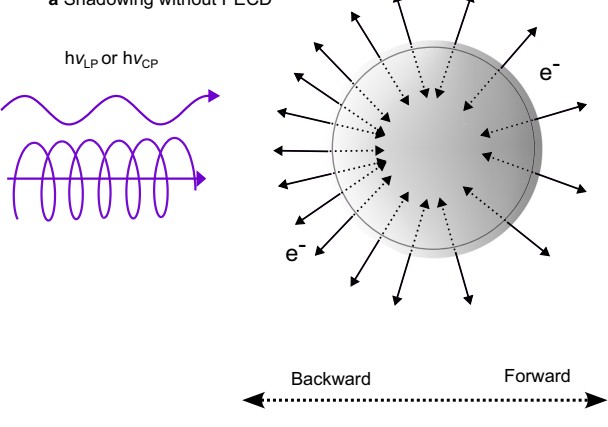

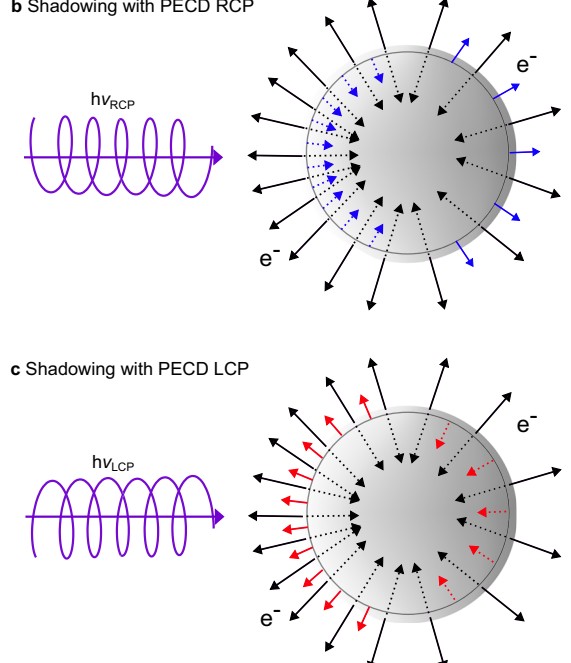

**Fig. 1 | Schematic representation of shadowing and PECD in nanoparticles.** **a** Shadowing in the absence of PECD is caused by the attenuation of light, of any polarization, within the nanoparticle. Isotropic creation of photoelectrons with an inhomogeneous spatial distribution creates an anisotropic electron image since only electrons emitted in an outward direction (full arrows) can escape the particle, while electrons emitted inward (dotted arrows) are reabsorbed in the nanoparticles. **b, c** PECD creates photoelectrons with a preference for initial momenta in the forward (blue) or backward (red) directions, indicated by the additional red and blue arrows. In combination with the preference for electron emission in the backward direction created from the shadowing effect, this leads to a net asymmetry of the photoemission yield for different handedness of the light polarization. More free photoelectrons are created if the PECD favors emission in the backward direction (**c**, LCP, $b_1^{+1} < 0$), as seen in this example by the more numerous red arrows pointing outward vs blue ones in (**b**).

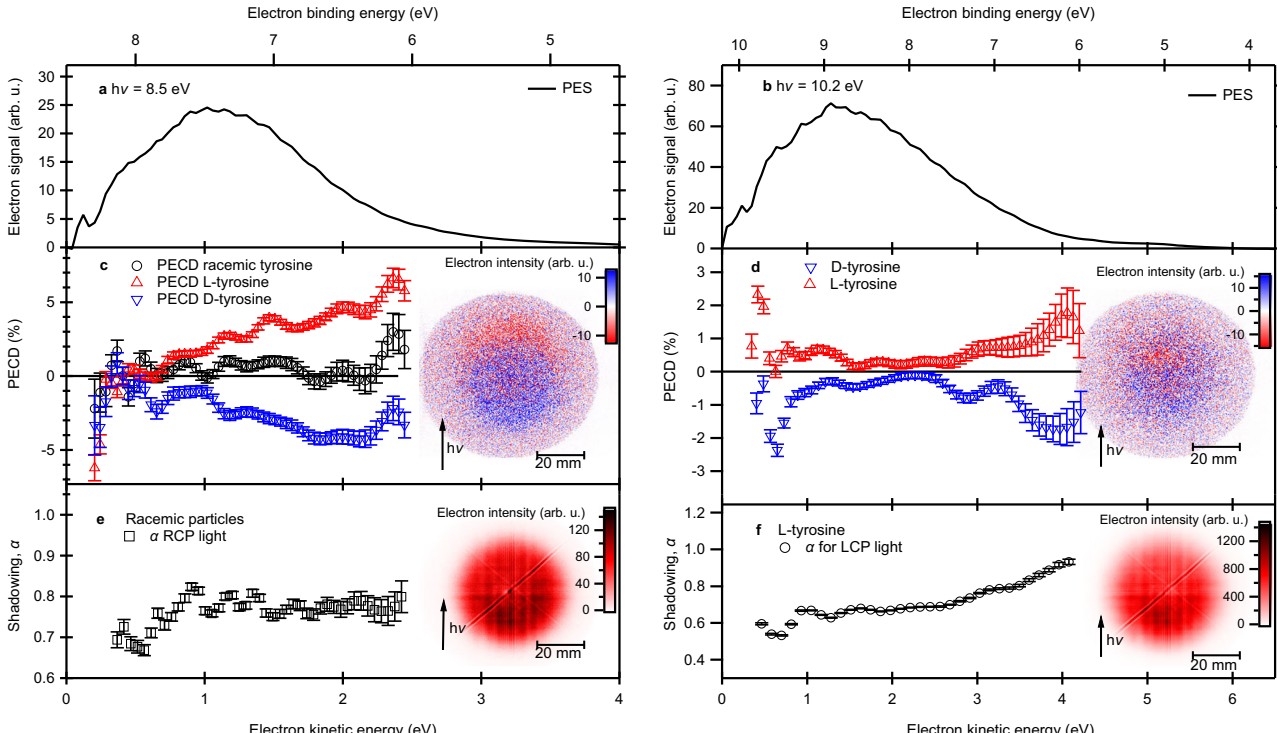

**Fig. 2 | Photoelectron imaging results for enantiopure and racemic tyrosine particles.** Comparison of ionization by VUV radiation of 8.5 eV (**a**, **c**, **e**) and 10.2 eV (**b**, **d**, **f**). Panels (**a**, **b**) show photoelectron spectra, panels (**c**, **d**) show the PECD asymmetry retrieved from the differences between images recorded with LCP and RCP light that are shown as insets. Panels (**e**, **f**) show the $\alpha$ parameter quantifying the shadowing effect. For better visibility (**e**, **f**), show only data for LCP. Data for RCP light is identical. Insets in panels (**e**, **f**) show the raw photoelectron images and the shadowing effect. The light propagation direction in the insets corresponds to the vertical direction, as indicated by the arrow and the spatial dimension of the images is indicated by the scale bar. Error bars in panels (**c**–**f**) correspond to the statistical standard deviations (see "Methods"). Source data are provided as a Source Data file.

photoemission yield reached a spectacular 10%[41]. While asymmetries of such magnitude are sometimes observed in PECD studies, they have not been observed in CD effects relying on the weaker E1M1 term in the expansion of the light matter interaction Hamiltonian. The surprising strength of the effect was attributed by these authors to the chiral crystal lattice formed by tyrosine: the chiral assembly of aromatic rings in the crystal lattice creates a supramolecular chirality[42] that could enhance the chiroptical asymmetries.

Motivated by these remarkable chiroptical properties, here, we present a photoelectron imaging study of enantiopure and racemic tyrosine nanoparticles, and discuss the interplay between the chiral asymmetry created by the PECD effect and the nanoparticle-specific shadowing effect. The combination of these two effects offers an alternative explanation for the previous observations and may be used as a basis for an easy-to-implement, affordable table-top set-up dedicated to chiral analysis of fragile biomolecular samples and aerosols.

## Results

### Photoelectron imaging of tyrosine nanoparticles

The results of this study on tyrosine (Tyr), performed at the DESIRS beamline of synchrotron SOLEIL in St. Aubin, France[43–45] are summarized in Fig. 2. The photoemission spectra, recorded with photon energies of 8.5 and 10.2 eV (Fig. 2a, b), show a gradual rise starting from an electron binding energy eBE = $h\nu - E_{\text{kin}} \approx 5$ eV indicating a drastic reduction from the gas phase ionization energy of 8.0 eV[46–48] upon condensation. The photoelectron signals fall of sharply toward zero kinetic energy, which has been observed previously for other nano-particles and can be assigned to inelastic electron scattering processes and recombination of the slowest photoelectrons[6,37]. The shadowing effect in the photoelectron images shown in the insets in Fig. 2e, f is

clearly visible. The angular distribution of shadowed photoelectron images can be considered in the form[37]

$$I(\theta) = b_0(1 + b_1^p \cos\theta + b_2^p(3\cos^2\theta - 1))(1 + b_\alpha(1 + \cos\theta)), \quad (2)$$

where

$$b_\alpha = \frac{\pi(1-\alpha)}{\alpha(\pi - 2) - (\pi + 2)} \leq 0 \quad (3)$$

and $\alpha$ is the shadowing parameter, displayed in the in Fig. 2e, f and taking mean values of 0.65 at 10.2 eV and 0.75 at 8.5 eV. Note that the $\alpha$ values depend sensitively on the size and refractive index of the aerosol particles at the used radiation wavelength, because both directly influence the induced light intensity distribution within the particle. However, there is also a weak dependence of $\alpha$ on the electron kinetic energy due to energy-dependent electron transport properties[35–37]. The chiral asymmetry due to the PECD in these particles can be observed in the difference between two normalized electron images obtained for left- and right-handed CPL displayed in the insets in Fig. 2c, d for L-tyrosine. Although the angular distribution of this difference image $\Delta I$ deviates slightly from the pure cosine shape expected in the case without shadowing (compare Eqs. (1) and (2)), the magnitude of the PECD effect can still be determined quantitatively as[6]

$$\text{PECD} = \Delta I^{\text{backward}} - \Delta I^{\text{forward}} = \frac{4\alpha}{(1+\alpha)^2}\frac{2b_1^{\pm 1}}{1+b_2^{\pm 1}} \approx 2b_1^{\pm 1}. \quad (4)$$

Note that the approximation above neglects minor contributions of the shadowing parameter and the anisotropy parameter. The effect

of the former arises from the normalization step (see below), and results in an underestimation of $b_1$ by our case, less than 5% and the latter is assumed to be zero. The obtained PECD values as a function of electron energy are given for both enantiomers as red and blue symbols in the middle panels of Fig. 2. At 8.5 eV an additional measurement of racemic nanoparticles (black symbols), clearly shows the absence of similar asymmetries within our statistical error bars (see "Methods"). The data obtained for L- and D-tyrosine nanoparticles show clearly the expected mirrored behavior, confirming that chirality is at the origin of the observed effect. The observed asymmetries reach values of 2% at a photon energy of 10.2 eV and 5% at 8.5 eV. For both measurements, the asymmetry is largest in the rising flank of high electron kinetic energies (electron binding energy $\approx 6$ eV), created by photoionization from the highest occupied molecular orbital of tyrosine. The observed asymmetries are larger than the asymmetries previously observed for solid nanoparticles of the amino acid serine[6], where the observed asymmetries did not exceed 1%. The comparison to previous PECD measurements on gas phase tyrosine[49], indicates that the PECD effect in nanoparticles is enhanced. This observation supports previous discussion of enhanced chiroptical effects by a chiral crystal lattice[41,42], creating a chiral supramolecular arrangement of tyrosine molecules. The observed asymmetries in the photoelectron angular distributions are on the same order of magnitude as the asymmetries observed in the total tyrosine nanoparticle photoemission yield[41].

### Interplay between PECD and nanoparticle shadowing

From ref. 6 and the present study, it is clear that the quantitative measurement of PECD signatures of solid nanoparticles is possible. A weak interplay between PECD and the shadowing anisotropy was already addressed in ref. 6 and leads to the inclusion of the shadowing parameter $\alpha$ in Eq. (4). This effect is more pronounced for stronger shadowing (smaller $\alpha$) but does not exceed a few percent of the PECD ($< 5\%$ for $\alpha \geq 0.65$) and is therefore much smaller than the statistical error bars of the PECD. A potentially more drastic interplay between shadowing and PECD effects has so far been overlooked, and is schematically explained in Fig. 1b. On top of the anisotropic photoemission due to shadowing, indicated by the density of black arrows, the PECD effect adds a preference for photoemission in the forward or backward direction, indicated by additional blue or red arrows, respectively. The latter depends on the choice of enantiomer and light helicity. The combined effect leads to a net variation of the yield of photoelectrons emitted from the particle, indicated by the total number of outward-pointing arrows, with the choice of enantiomer or light helicity. If the PECD favors photoemission in the forward direction (Fig. 1b, $b_1 > 0$), the number of produced photoelectrons will be lower than in the case of PECD favoring photoemission in the backward direction (Fig. 1c, $b_1 < 0$). This effect corresponds phenomenologically to a circular dichroism in the photoemission cross section, but unlike electronic CD, it is described in the pure electric dipole approximation and is therefore orders of magnitude more intense[38,39]. In other words, PECD mediated by the shadowing effect leads to an apparent CD in the total integrated electron yield that we refer to as chiral asymmetry of the photoemission yield, to avoid confusion with traditional CD that requires magnetic dipole contributions.

To assess CAPY more quantitatively, we consider the total photoemission yield obtained by integration of Eq. (2), for a given electron energy and for $b_2 \approx 0$, which is often found in aerosol photoemission[6,50], and assuming a negligible absorption CD

$$Y = \pi b_0 \left(1 + b_\alpha \left(1 + \frac{b_1^p}{2}\right)\right). \tag{5}$$

The photoemission yield depends both on the shadowing and chirality of the condensed system, in contrast to the total (angle-integrated) yield obtained from Eq. (1), i.e., $Y_m = \pi b_0$. Obviously, for

achiral or racemic systems ($b_1^p = 0$) the total yield of a condensed system $Y_0$ is attenuated by a factor $(1 + b_\alpha)$, since $b_\alpha < 0$. For chiral systems in which $b_1^p < 0$, corresponding to preferred backward emission, the attenuation factor will be smaller, resulting in an increase in the number of produced electrons ($Y_- > Y_0$). Conversely, when $b_1^p > 0$ (i.e., preferred forward emission), the attenuation factor becomes larger, and the total yield is lower ($Y_+ < Y_0$). Ignoring this expected difference in the photoemission yield in the normalization step for the extraction of the PECD is at the origin of the above-mentioned weak dependence of the PECD on $\alpha$ (Eq. (4)). Calculated dependences of $Y_+$ and $Y_-$ total yields, normalized to the total yield of an achiral system $Y_0$, on the shadowing parameter $\alpha$ for $|b_1^p| = 0.05$ are presented in Fig. 3, alongside the Kuhn-type asymmetry parameter

$$g^{E1} = 2 \frac{Y_- - Y_+}{Y_- + Y_+} = \frac{b_\alpha b_1^-}{(1 + b_\alpha)} \tag{6}$$

used to quantify the CAPY arising strictly from electric dipole effects (E1).

In our synchrotron-based measurements, this effect cannot be directly observed due to experimental fluctuations in sample density and photon flux, which are faster than the relatively slow light helicity switch that can reasonably be achieved with the VUV undulator ($\sim 10^{-3}$ Hz corresponding to a $\approx 95\%$ duty cycle), leading to error bars of a few percent for integral measurements.

### Quantitative estimation of CAPY

We will, in the following, provide a quantitative estimate of the CAPY expected for tyrosine nanoparticles under our experimental conditions, and compare the expected size dependence of the asymmetries to previous observations reported as CD effect ($g^{E1M1}$)[41]. In addition, we will discuss the range of nanoparticle properties for which we expect an observable CAPY effect, and discuss an experimental setup and procedure to use CAPY as an analytical approach with high sensitivity to chirality.

As a first step, we characterize the light intensity distribution within the tyrosine nanoparticles that are necessary to produce the observed shadowing effects, which is independent of the polarization of the ionizing radiation. In this characterization, tyrosine

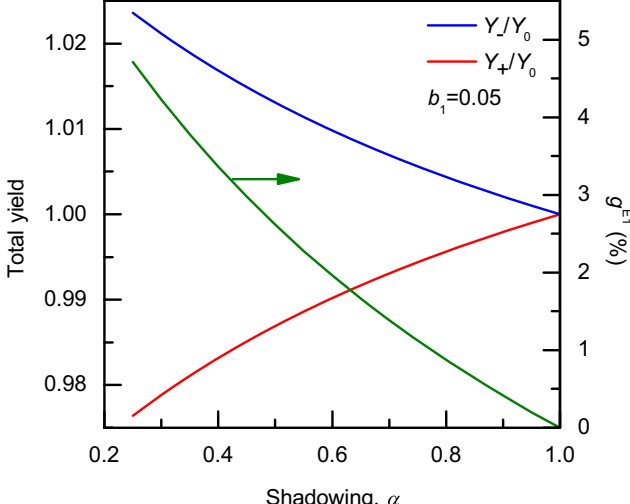

**Fig. 3 | Influence of the particle shadowing on CAPY.** Photoemission yield, calculated using Eq. (5), for chiral nanoparticles interacting with circularly polarized light with a chiral anisotropy parameter of $b_1^p = 0.05$ (red line) and $b_1^p = -0.05$ (blue line), as a function of the achiral shadowing parameter $\alpha$. The resulting asymmetry parameter CAPY quantified by $g^{E1}$ (Eq. (6)) for the selected $b_1^{\pm 1}$ are shown as a green line on the right axis. Source data are provided as a Source Data file.

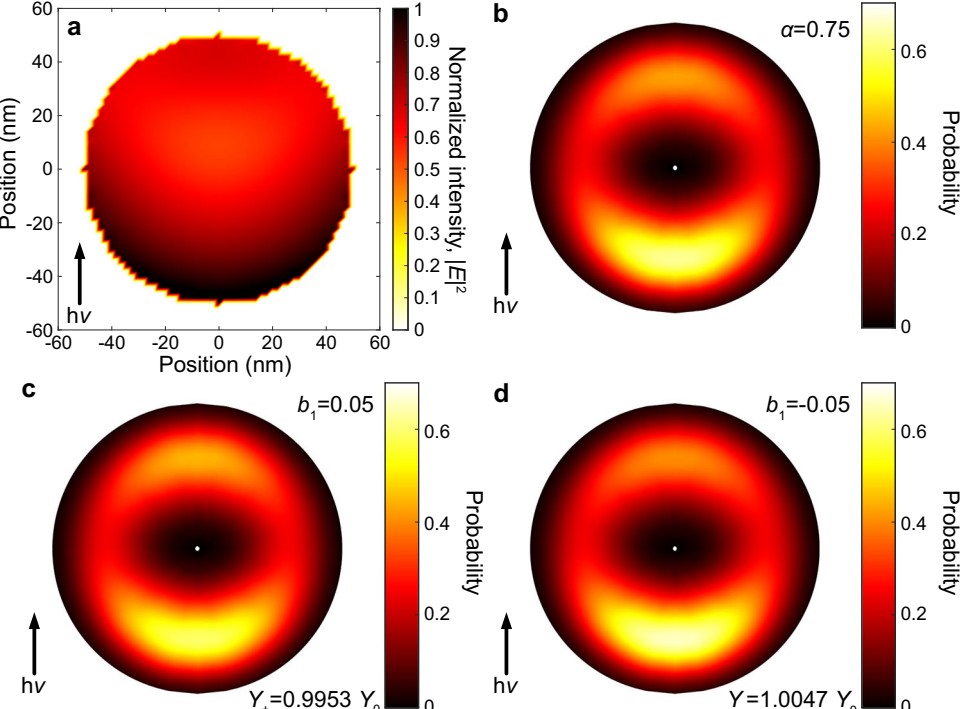

**Fig. 4 | Quantitive estimation of CAPY in tyrosine nanoparticles. a** Distribution of the light intensity (square of electric field amplitude) within a spherical particle 100 nm in size of refractive index 1.3 + 0.5i obtained from a DDA calculation for 8.5 eV photon energy. **b–d** Photoelectron angular distributions in momentum space corresponding to the light intensity distribution shown in panel (**a**) for particle diameters of 100 nm and an electron escape length of 5 nm. The absolute scale of the momenta along vertical and horizontal axes of the photoelectron angular distributions in panels (**b**–**d**) are arbitrary, and only the angular distribution is of relevance. **b** simulated angular distribution for $b_1^p = 0$ (achiral case), **c** $b_1^p = 0.05$ and **d** $b_1^p = -0.05$. The total photoionization yields for $b_1^p = \pm 0.05$ are given as labels in panels (**c**, **d**). The arrows indicate the direction of the propagation of the incident radiation. Source data are provided as a Source Data file.

nanoparticles are considered spherical with a mean diameter of 100 nm, which is a typical size of the nanoparticles produced by the aerosol setup at DESIRS. The light intensity (square of electric field amplitude) distribution within the particle was calculated using a discrete dipole approximation (DDA)[51] for an estimated complex value of the refractive index. Subsequently, the electron angular distribution is simulated assuming photoelectrons are created with an isotropic momentum distribution within the particles, but can only escape from a few-nanometer thick surface layer without changing their initial momenta. Subsequently, the $\alpha$ parameter is determined for the photoelectron angular distribution. The complex refractive index was varied in an iterative way until $\alpha$ agreed with the experimentally observed values. The best matches were obtained using refractive indices $m = n + ik$ of 1.3 + i0.5 for 8.5 eV and 1.3 + i0.6 for 10.2 eV. The used values for the extinction coefficient $k$ reasonably agree with the reported value of 0.389 for tyrosine thin films at 5.9 eV[52]. In this way, we obtain estimates of the complex refractive indices for tyrosine nanoparticles. The resulting light intensity distribution and simulated photoelectron image for 8.5 eV are shown in Fig. 4a and b. Note that the choice of the shape of the photoelectron kinetic energy distribution for the electron image is arbitrary, and does not affect the resulting angular distribution, since details of possible energy-dependent electron scattering processes are not explicitly treated in this model.

In a second step, we assume the photoelectrons to be created following the same spatial light intensity distribution, while we additionally include the superimposed preference for forward or backward emission created by the PECD effect. Resulting photoelectron images for LCP and RCP light, obtained for values of $b_1^p = \pm 0.05$ are shown in Fig. 4c, d. The slight differences in the corresponding images can be observed. Less clearly visible is the difference in electron yield, given here as the probability for photoelectrons to escape from the particle

in the image labels. The values are given here relative to the value of $Y_0$ corresponding to the shadowed image without PECD of Fig. 4b. The difference between the value of $Y_+ = 0.9953\, Y_0$ obtained for $b_1^p > 0$ (indicating preference for the emission in the forward direction) and $Y_- = 1.0047\, Y_0$ for $b_1^p < 0$ (backward direction) amounts to a factor of $g^{E1} = 2\frac{Y_- - Y_+}{Y_- + Y_+} = 0.0094$. This value agrees very well with the prediction of Eq. (6) depicted in Fig. 3, and serves as an independent confirmation of our mathematical description (see detailed comparison in Supplementary Fig. 2). Furthermore, this value is slightly below the range of $g$ values between 2%–10% reported for tyrosine particles in ref. 41, but higher than typical values of UV circular dichroism reported for electronic transitions usually lying in the few 0.001% and 0.1% range[38].

To analyze the particle size dependence of the observed asymmetry in the photoionization yield, repeated simulations with the above values of the complex refractive index (1.3 + i0.5 for h$\nu$ = 8.5 eV and 1.3 + i0.6 for h$\nu$ = 10.2 eV) and various sizes of nanoparticles were performed. The results are summarized for photon energies of 8.5 eV and 10.2 eV in Fig. 5. Both cases show a decreasing value of $\alpha$ (bottom panels), i.e., stronger shadowing, with larger particle size. In agreement with Eq. (6) and Fig. 3 this leads to a stronger asymmetry of the photoemission yield for larger particles. It is interesting to note that the resulting trend of $g^{E1}$ as a function of particle size closely resembles the particle size dependence observed previously[41]. Quantitatively, the simulations for 8.5 eV photon energy and a $b_1 = 0.025$, corresponding to our experimental observations, predict a $g^{E1}$ of about 0.5% for 100 nm particles. For larger particles, it increases to about 2%, which is very similar in behavior to the asymmetries observed previously. To obtain a CAPY close to the maximum asymmetry reported in ref. 41 $|b_1| > 0.05$ would be necessary, assuming that the shadowing effect behaves similarly at 6.43 eV, the photon energy used. Additional observations in ref. 41, like the increase of the asymmetry with decreasing light intensity and an increase of the asymmetry for more

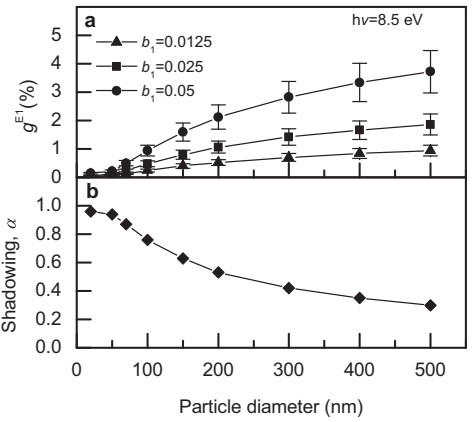
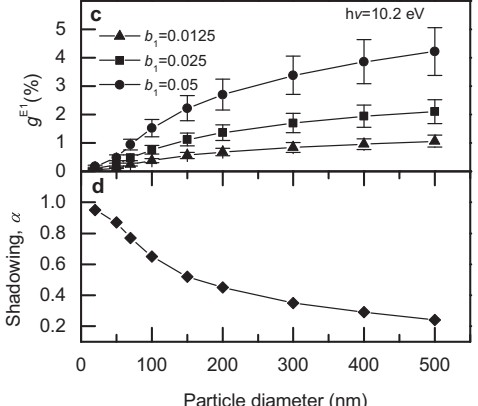

**Fig. 5 | Analysis of the size dependence on CAPY. a** Simulated size-dependent chiral asymmetries in the photoionization yields $g^{E1}$ for 8.5 eV photon energy for $|b_1^p|$ equal to 0.0125, 0.025, and 0.05. **b** The dependence of shadowing parameter $\alpha$ on the diameter of aerosol particles for 8.5 eV photon energy. Corresponding results obtained for 10.2 eV photon energy are presented in panels (**c**, **d**). The error bars of $g^{E1}$ were estimated to $\approx 20\%$ of the values based on the uncertainties due to finite variable sampling in the simulations and the electron escape length value used. Source data are provided as a Source Data file.

crystalline particles, may as well be explained based on the CAPY effect described here (see detailed discussion in Supplementary Note 2). It seems thus reasonable to assume the previously observed asymmetry in the photoemission yield is explained by the chiral asymmetry in photoemission created by PECD combined with shadowing within the electric dipole approximation, rather than by a strong enhancement by several orders of magnitude of traditional (E1M1) CD effects.

## Discussion

The possible use of PECD in an analytical context, with its large associated asymmetries and exquisite sensitivity well-suited for dilute matter, has been a strong driving force for the development of PECD studies since the very beginning. The use of laser sources for PECD studies in the last decade has enabled a significant step in this direction, enabling table-top experiments not requiring large-scale facilities like synchrotron light sources[53,54]. So far, however, with the exception of a PECD study on anions produced by electrospray[55], real-time in-situ determination of enantiomeric excess has been performed exclusively on thermally stable model systems allowing decomposition-free vaporization[56–58].

Gas-phase PECD studies of biologically active sample systems of high interest remain challenging, largely due to the difficulty in producing dense targets. Generation of unsupported nanoparticles of thermally fragile but soluble molecules is on the contrary, straightforward. Thus, the described effect opens a direct pathway to the detection of PECD effects via the chiral asymmetry of the photoemission yield, promising to be a valuable and universal analytical tool for chiral biomolecules and pharmacologically active substances with no requirements in terms of thermal stability upon vaporization. Moreover, the generally low ionization energies of these particles fall within the UV range and thus are easily accessible with laboratory-based light sources. Note that the information obtained from CAPY measurements is not equivalent to Gas-phase PECD measurements. Condensation to particulate form is expected to affect the molecular conformer distribution as well as possibly tautomer population, while the molecular environment can affect the observed PECD as well. Because neither the tautomer and conformer population nor the molecular environment affects the enantiomeric composition, the PECD induced CAPY effect can be exploited in ex-situ measurements to quantify enantiomeric excess of samples, as a potential alternative to solution phase CD measurements or enantioselective chromatography methods.

The described effect of a chiral asymmetry in the photoemission yield created by PECD in conjunction with particle shadowing is more easily measured than PECD using electron imaging techniques or electrostatic electron analyzers, requiring a high vacuum environment. An experimental apparatus for such measurements, following previous designs[40,41], is shown in Fig. 6. It does not require extensive vacuum systems or detection technologies. It requires exclusively tools for the creation of nanoparticles from solution and their characterization, as well as UV light sources and their polarization control, in a simple one-photon excitation scheme. Designs using UV lamps instead of laser sources are conceivable as well. Upon characterization of the isolated enantiopure samples, such measurement can be easily used as an analytical procedure with high sensitivity to enantiopurity.

In addition, this rather simple setup could be used to study enantiomeric excess in pre-formed aerosol particles in environmental, biomedical, or catalytic applications. For instance, it could be combined with atmospheric aerosol inlet devices and used as an analytical tool for the in situ analysis of chiral organic aerosols. Such aerosols are present in the earth's atmosphere in the form of secondary organic aerosols[59,60] whose real-time in situ chiroptical response might be a tracer of changes in volatile emissions of forests[61]. Also, in the food, pharmaceutical and chemical industries, where often solid products in the form of powders of well-defined granularity are produced by spray-drying liquid solutions, CAPY could be employed as an in-situ observable for quality control monitoring directly in the production line.

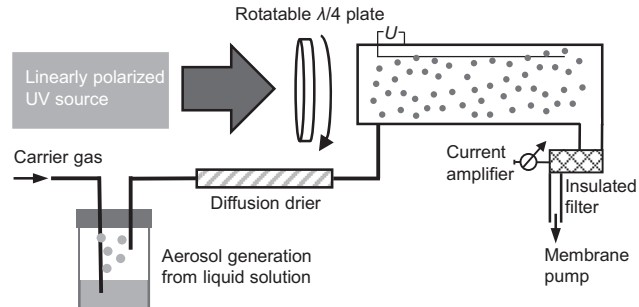

**Fig. 6 | Proposed setup for measurements of CAPY.** Setup following ref. 40. Aerosol droplets are created from liquid solutions and subsequently dried and ionized by circularly polarized UV radiation of variable helicity. Electrons and light ions are separated by an electric field from the heavy positively charged aerosol particles, that create a measurable current. The current changes in magnitude depend on the light helicity, allowing to quantitatively determine the CAPY.

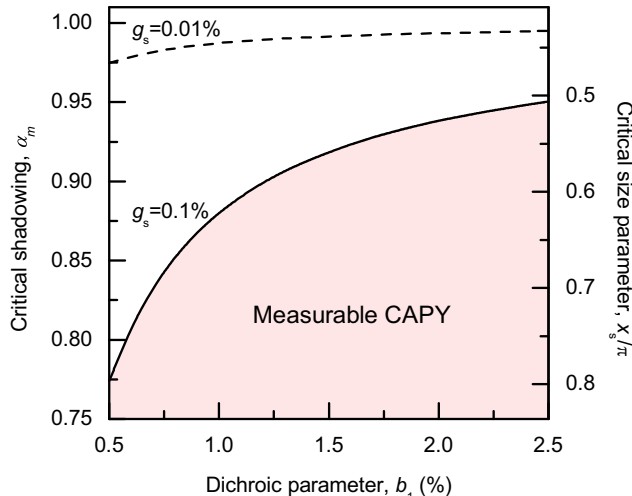

**Fig. 7 | Estimation of the applicability of CAPY.** Critical values of the shadowing parameter $\alpha_m$ below which the CAPY is measurable in aerosols with dichroic parameter $b_1$ and anisotropy factor $g_s$. The corresponding critical values of the size parameter $x_s = |m| \pi D/\lambda$ are also presented. Source data are provided as a Source Data file.

Enantiomeric excess in thus formed particles can be used to assess the impact of drying process and protective agents on structural and functional changes in protein components[62,63]. Another direct application for the analysis of particulate matter is conceivable for the characterization of chiral nanosystems, including nanostructures of chiral morphology[64] and nanoparticles functionalized by chiral molecules[65]. Similar PECD induced CAPY effects should be observable for photoemission from thin molecular films, liquid jets or surfaces, as these environments create very strong or total shadowing effects. The observation of CAPY from surfaces may be less straightforward than in the nanoparticle case, but the exploitation of the effect may provide a pathway towards the analysis of chiral functionalized surfaces applied for asymmetric catalysis.

The sensitivity of measurements of the chiral asymmetries of the photoemission yield have been reported around $g_s = 0.01\%$[40], but can likely be improved by faster switching of the light polarization and other technological advances. We propose that a quantitative determination of enantiomeric excess in a sample with the current sensitivity requires somewhat larger asymmetries of $g_s = 0.1\%$ for the enantiopure samples. Using these two values we estimated the values of critical shadowing parameter parameters $\alpha_m$ for a given range of $b_1$ that are required to experimentally measure a CAPY or to use the CAPY for a quantitative analysis of the enantiopurity of the sample, based on Eq. (6). The results are graphically represented in Fig. 7. To obtain a quantitatively usable CAPY effect exceeding 0.1% for a small $b_1 \approx 1\%$, only a weak shadowing with $\alpha_m \lesssim 0.88$ is necessary, while for $b_1 \approx 2.5\%$ aerosol particles of almost any material should demonstrate measurable CAPY upon absorption of radiation of wavelength $\lambda$ above the ionization potential. The $\alpha_m$ values could be used to determine the corresponding critical size parameters $x_s = |m| \pi D/\lambda$, also shown in Fig. 7. Based on these values, we estimate that CAPY should be quantitatively usable for $|m|D \geq 0.5\lambda$, where $|m|$ is the absolute value of the complex refractive index. This condition is usually satisfied for organic aerosol particles of chiral compounds between 100 and 500 nm in diameter near the ionization threshold ($\approx 200$ nm).

While our results and analysis focus on the UV and VUV range, similar measurements may also be feasible through the visible and IR spectral range at sufficiently high light intensities, relying on multiphoton ionization processes. Depending on the complex refractive

indexes of the sample materials at any given photon energy, different anisotropic light intensity distributions governed by nanofocussing rather than shadowing effects may be exploited[36,66,67]. The multiphoton character of the ionization process further increases the spatial anisotropy of the photoemission probability distribution within the particles, and hence increases the effect.

## Methods
### Experimental procedure
The experiments were performed employing the DELICIOUS III spectrometer[43,44] located at the DESIRS beamline[45] of the French synchrotron facility SOLEIL. Nanoparticles are created by spraying a 1 g L⁻¹ aqueous solution of D- or L-tyrosine (Sigma Aldrich, 98% purity) or their racemic mixture using an atomizer (TSI, model 3076) operated with 1.5 bar of helium. The nanoparticles are passed through diffusion driers (TSI, model 3062) before being focused into the SAPHIRS molecular beam chamber by an aerodynamics lens system[37]. After passing through a set of skimmers, the nanoparticles were ionized by circularly polarized VUV radiation of 10.2 and 8.5 eV from the monochromatized branch of the DESIRS beamline. Photoelectrons were detected on the dedicated Velocity Map Imaging detector of the DELICIOUS III spectrometer, while ions were not detected. Background signals were recorded while operating the nanoparticle source with deionized water for a shorter acquisition time than the measurements. Before subtraction, the background images were scaled in intensity accordingly to compensate for the shorter acquisition times. The resulting electron images were binned and converted to electron kinetic energy and angular distributions using the pBasex[37,68] algorithm. Error bars (standard deviations) given for the shadowing parameters $\alpha$ or the magnitude of PECD ($b_1$) are purely statistical and assume a Poissonian counting statistics on each pixel of the electron image. Absolute systematic uncertainties, especially on the shadowing parameter $\alpha$ arising for example from uncertainties in the determination of the image center, may be larger than the statistical errorbars. The DESIRS beamline provides circularly polarized light of both helicities with a high purity of 97%–99%, but the photon flux varies by a few percent between the two helicities. The latter does not affect measurements of the PECD, which is determined from the difference of normalized images recorded with the different light helicities[22], but makes direct measurements of circular dichroism effects of the photoionization yield challenging.

The morphology of tyrosine aerosols, deposited onto a conductive substrate using a Nanometer Aerosol Sampler (TSI model 3089), was investigated by a JEOL JSM-6390 LV scanning electron microscope (SEM) operating at 15 kV.

### Simulations
The distribution of the square of the electric field amplitude for a spherical target, representing average of all aerosol particles traversing the interaction region, in a plane containing the direction of the VUV radiation propagation ($|E(x, y, z = 0)|^2$) was calculated using DDA with the DDSCAT code[51]. The $|E|^2$ distribution depends sensitively on the target particle size ($D$) and complex refractive index of the particle at given photon energy h$\nu$. From the $|E|^2$ distribution, photoemission probability $P_0$ was calculated using[35,37]

$$P_0(E_k, \theta; h\nu, D) = \sum_{x,y} |E(x,y)|^2 B(E_k, D) e^{-\mu d} \Theta(\xi_c - \xi), \quad (7)$$

where $d = d(x, y, \theta)$ is the distance that the spawned photoelectron traverses from its initial location within the particle $(x, y)$ in the $\theta$ direction towards the vacuum, $\mu$ is the electron escape length, $B(E_k, D)$ is the transmission function[37], and $\Theta(\xi_c - \xi)$ is the step function which limits the range of angles perpendicular to the particle surface $\xi$ into which electron can escape[37]. From Eq. (7), the shadowing parameter $\alpha$

and the total yield of the shadowed image $Y_0$ were calculated as

$$\alpha = \frac{\sum_{\theta=0}^{\pi/2}\sum_{E_k=0}^{E_k^{max}}P_0(E_k,\theta)n_d(\bar{E_k},\sigma)}{\sum_{\theta=\pi/2}^{\pi}\sum_{E_k=0}^{E_k^{max}}P_0(E_k,\theta)n_d(\bar{E_k},\sigma)} \text{ and} \quad (8)$$

$$Y_0 = \sum_{\theta=0}^{\pi}\sum_{E_k=0}^{E_k^{max}}P_0(E_k,\theta)n_d(\bar{E_k},\sigma), \quad (9)$$

where $n_d$ is normalized electron energy distribution, which was assumed to be Gaussian with mean and FWHM equal to $(\bar{E_k},\sigma) = (1\,\text{eV}, 1\,\text{eV})$ for $h\nu = 8.5\,\text{eV}$ and $(\bar{E_k},\sigma) = (1.2\,\text{eV}, 1\,\text{eV})$ for $h\nu = 10.2\,\text{eV}$ to match the experimental PES spectrum. To include the PECD effects in the simulation, the asymmetric photoemission probabilities $P_+$ and $P_-$ were calculated using Eq. (10),

$$P_\pm(b_1,E_k,\theta;h\nu,D) = P_0(E_k,\theta;h\nu,D)(1\pm b_1\cos\theta), \quad (10)$$

with the total yields $Y_+$ and $Y_-$ also given by Eq. (5) using appropriate photoemission probability. The chiral asymmetry in the photoionization yield ($g^{E1}$) was calculated using

$$g^{E1} = 2\frac{Y_{LCP}-Y_{RCP}}{Y_{LCP}+Y_{RCP}} = 2\frac{Y_- - Y_+}{Y_- + Y_+}, \quad (11)$$

considering that for L-amino acid molecules under the LCP parameter $b_1$ is negative[6,28,31].

### Reporting summary

Further information on research design is available in the Nature Portfolio Reporting Summary linked to this article.

## Data availability

The data that support the findings of this study, including unprocessed raw data, are available from Figshare[69] and from the corresponding author upon request. Source data are provided in this paper.

## Code availability

The code used to produce the data in Figs. 4, 5, and 7 are available from Figshare[69] and from the corresponding author upon request.

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

## Acknowledgements

We thank Jean-Francois Gil for his technical help around the molecular beam chamber. Furthermore, we are grateful to the general SOLEIL staff for smoothly running the facility, especially under project 20170100. S.H. thanks the DFG for support in the framework of RTG 2717, and through the grants HA 10463/2-1 and HA 10463/3-1. D.K.B. acknowledges support of the Ministry of Science, Technological Development and Innovation of the Republic of Serbia under Grant Agreements No 451-03-33/2026-03/ 200017. L.N and G.A.G acknowledge support by state funding from the ANR under the France 2030 program, with reference ANR-23-EXLU-0004, PEPR LUMA TORNADO.

## Author contributions

S.H. and L.N. conceived the research. S.H., G.A.G., and L.N conducted experimental measurements. S.H. evaluated the experimental data. D.K.B. performed the simulations and calculations. All authors discussed and interpreted the results. S.H. and D.K.B. visualized the data and wrote the initial draft of the manuscript, and all authors reviewed and edited the manuscript.

## Funding

## Competing interests

The authors declare no competing interests.
