## [Transparent Peer Review file · Nature Communications]

Linking photoelectron circular dichroism to the asymmetric total photoemission yield measured in aerosol nanoparticles of tyrosine

Corresponding Author: Dr Sebastian Hartweg

Version 0:

Reviewer comments:

Reviewer #1

(Remarks to the Author)

The manuscript reports theoretical and experimental study of angular distributions of photoelectrons emitted from free chiral nanoparticles irradiated with circularly polarized light. The study combines two known effects, PECD which emerges owing to the target's chirality, and shadowing effect owing to the finite nanoparticle's size. The latter effect differentiates between the "light" and "dark" sides of the nanoparticle and results in a preferable photoelectron emission in the backward direction wrt. the light propagation direction. Both the PECD and the shadowing effects are observed in the current study. However, such an achievement is already documented in the previous work of the authors in Ref. [6].

As the next step, the authors elaborate theoretically, that a combination of these two effects must result in a Chiral Asymmetry in the total Photoemission Yield. The effect should be distinguished from the conventional CD, since it is governed by the PECD which relies on the electric dipole approximation. Therefore, it is predicted to be large (on the order of a few per cent), as follows from the quantification and exemplification of the CAPY estimated in the manuscript. I must say I, was really impressed by this expected effect, its simplicity, accessibility, and a broad range of potential applications, as discussed by the authors. I think that, even though it is only predicted theoretically and not demonstrated experimentally here, this work deserves publication in Nat. Commun. I just have a few suggestions for improvement.

I was slightly confused by understanding equation (2) when reading the ms. Only retrieving to the literature made the point clear for me. In particular: The authors explanation on page 6 (below equation 5) suggests that fluctuations in the sample density and photon flux do not allow to access here the integral property (total electron yield). This point is further elaborated at the end of the Experimental Methods section. It is stated as well that such fluctuation do not affect the PECD measurements, since it is obtained as a difference between the NORMALIZED images measured with different light helicities (before Eq (2) and in the Experimental Methods section). This, in turn, contradicts Eq. (2) which uses one light helicity for the extraction of the PECD at the angles 0 and 180. Moreover, the procedure described by the authors is standard for a conventional PECD where the total electron yields (under equal conditions) is independent of the light helicity. Even one cannot achieve those equal conditions experimentally, such a normalization of the images prior their subtraction "repairs" the measurement. Now, in the case considered here, the total electron yields are explicitly different for two light helicities, as given by Eqs. (3) and (4), which appear two pages later. In addition, the mentioned above inconsistency around Eq. (2) can be understood only when looking in the SM of the previous work of the authors [6], where the shadowing effect was quantified and Eq. (2), which uses two emission angles for one helicity, was derived. I would propose to repair this loop in the discussion such that Eqs. (3) emerge before Eq. (2) and a clear difference wrt. Eq. (1) is discussed first. Thereafter, Eq. (2) appears in another form, as is spelled in text, and, if necessary, a transformation to the current form appears.

Minor points:

- in Eq. (2): parameter b_2 is not defined. It is either $b_2^{\{+1/-1\}}$ or $-1/2*b_2^0$.
- In Fig. 3: +/- subscripts in the legend are unreadable.

Reviewer #2

(Remarks to the Author)

The article "Large circular dichroism in the total photoemission yield of free chiral nanoparticles created by a pure electric dipole effect" presents a novel chiroptical analytical method, CAPY, chiral asymmetry of the photoionization yield. This new method, as the authors point out, can be used in experimental set-ups of relatively low complexity, creating low entry requirements a chiroptical technique that can be described within the electric dipole approximation. Given the large community of scientists interested in exploring chiral effect of isolated, but increasing large chiral molecules, we expect this work to attract a large audience. We also found the descriptions and proof-of-concepts within the article to be thorough. Due to the high practicability of the presented method and high quality of the work carried out, we believe the article meets the merit criteria for publication in Nature Communications pending very minor edits.

Some of the figures utilized formatting that made it difficult to interpret the figures. Here are suggested changes for the figures in the main text to improve interpretation ease:

- Fig. 1: The dotted arrows that indicate electrons emitted inwards should be more clearly dotted. I only noticed the dots after reading about it in the caption.
 - Fig. 2: The shapes for each data point in the graph are indiscernible. This is only really a problem in the bottom panel. The shapes used for LCP and RCP in the bottom panels are also reversed for the two different photon energies, as evident by the legend.
 - Fig. 3: The font size of the legend could be larger, to make the subscripts more easily readable. The fact that the green line shows the asymmetry parameter g should be added into the caption to improve clarity.
- We also recommend some minor changes to the text:
- Line 111: et al. should be italicized.
 - "Note that the α values depend sensitively on the size and refractive index of the aerosol particles at the used radiation wavelength, that directly influence the induced light intensity distribution within the particle."

Line 138: it is not clear to me what subject is providing the influence: the wavelength, or the size and refractive index. If it is only the wavelength, then it should be "influences".

- "As a first step we characterize the light intensity distribution within the tyrosine nanoparticles that are necessary to produce the observed shadowing effects; whatever is the polarization of the ionizing radiation."

Line 242: the second part of the sentence is confusing and should be rephrased.

- Finally, centering the equations on page 6 might help with the reading flow.

Reviewer #3

(Remarks to the Author)

Version 1:

Reviewer comments:

Reviewer #1

(Remarks to the Author)

The authors made appropriate efforts and changes wrt my comments and those of the second referee(s). The revised manuscript can now be accepted.

Reviewer #2

(Remarks to the Author)

I appreciate the changes the authors made to the manuscript and support the publication of the article in it's current state.

Reviewer #3

(Remarks to the Author)

Dear Reviewers

We thank you for your time and effort spent on thoroughly reviewing our manuscript and the positive evaluation of our work. We are grateful for your helpful comments aimed at improving the presentation of our work. We have followed all your suggestions and provide below a detailed list of the changes made to our manuscript during its revision.

Best regards on behalf of all coauthors

Sebastian Hartweg

In the list below your comments are given in blue and our reply in black.

Reviewer #1:

The manuscript reports theoretical and experimental study of angular distributions of photoelectrons emitted from free chiral nanoparticles irradiated with circularly polarized light. The study combines two known effects, PECD which emerges owing to the target's chirality, and shadowing effect owing to the finite nanoparticle's size. The latter effect differentiates between the "light" and "dark" sides of the nanoparticle and results in a preferable photoelectron emission in the backward direction wrt. the light propagation direction. Both the PECD and the shadowing effects are observed in the current study. However, such an achievement is already documented in the previous work of the authors in Ref. [6].

As the next step, the authors elaborate theoretically, that a combination of these two effects must result in a Chiral Asymmetry in the total Photoemission Yield. The effect should be distinguished from the conventional CD, since it is governed by the PECD which relies on the electric dipole approximation. Therefore, it is predicted to be large (on the order of a few per cent), as follows from the quantification and exemplification of the CAPY estimated in the manuscript. I must say I, was really impressed by this expected effect, its simplicity, accessibility, and a broad range of potential applications, as discussed by the authors. I think that, even though it is only predicted theoretically and not demonstrated experimentally here, this work deserves publication in Nat. Commun. I just have a few suggestions for improvement.

We thank the reviewer for this positive overall assessment.

I was slightly confused by understanding equation (2) when reading the ms. Only retrieving to the literature made the point clear for me. In particular: The authors explanation on page 6 (below equation 5) suggests that fluctuations in the sample density and photon flux do not allow to access here the integral property (total electron yield). This point is further elaborated at the end of the Experimental Methods section. It is stated as well that such fluctuation do not affect the PECD measurements, since it is obtained as a difference between the NORMALIZED images measured with different light helicities (before Eq (2) and in the Experimental Methods section). This, in turn, contradicts Eq. (2) which uses one light helicity for the extraction of the PECD at the angles 0 and 180. Moreover, the procedure described by the authors is standard for a conventional PECD where the total electron yields (under equal conditions) is independent of the light helicity. Even one cannot achieve those equal conditions experimentally, such a normalization of the images prior their subtraction "repairs" the measurement. Now, in the case considered here, the total electron yields are explicitly different for two light helicities, as given by Eqs. (3) and (4), which appear two pages later. In addition, the mentioned above inconsistency around Eq. (2) can be understood only when looking in the SM of the previous

work of the authors [6], where the shadowing effect was quantified and Eq. (2), which uses two emission angles for one helicity, was derived. I would propose to repair this loop in the discussion such that Eqs. (3) emerge before Eq. (2) and a clear difference wrt. Eq. (1) is discussed first. Thereafter, Eq. (2) appears in another form, as is spelled in text, and, if necessary, a transformation to the current form appears.

We thank the reviewer for pointing out the difficulty in understanding the original formulation in our discussion. We followed the recommendation and have slightly restructured the section, changing the ordering of the equations and adjusting former equation (2) to appear in terms of the difference image, rather than a single electron image. In addition, we have improved the text clarity, to help the reader to understand the differences between the shadowed and non-shadowed images.

Minor points:

*-in Eq. (2): parameter b_2 is not defined. It is either $b_2^{+1/-1}$ or $-1/2*b_2^0$.*

The reviewer is correct, and we have adjusted the equation using $b_2^{+1/-1}$.

-In Fig. 3: +/- subscripts in the legend are unreadable.

The reviewer is correct and we have adjusted the figure legends for better readability.

Reviewer #2:

The article "Large circular dichroism in the total photoemission yield of free chiral nanoparticles created by a pure electric dipole effect" presents a novel chiroptical analytical method, CAPY, chiral asymmetry of the photoionization yield. This new method, as the authors point out, can be used in experimental set-ups of relatively low complexity, creating low entry requirements a chiroptical technique that can be described within the electric dipole approximation. Given the large community of scientists interested in exploring chiral effect of isolated, but increasing large chiral molecules, we expect this work to attract a large audience. We also found the descriptions and proof-of-concepts within the article to be thorough. Due to the high practicability of the presented method and high quality of the work carried out, we believe the article meets the merit criteria for publication in Nature Communications pending very minor edits.

We thank reviewers 2 and 3 for this positive assessment.

Some of the figures utilized formatting that made it difficult to interpret the figures. Here are suggested changes for the figures in the main text to improve interpretation ease:

- Fig. 1: The dotted arrows that indicate electrons emitted inwards should be more clearly dotted. I only noticed the dots after reading about it in the caption.*

We thank the reviewers for pointing this out to us. We have adjusted the dotted arrows, and are prepared to do so again, should this be necessary in a final layout.

- Fig. 2: The shapes for each data point in the graph are indiscernible. This is only really a problem in the bottom panel. The shapes used for LCP and RCP in the bottom panels are also reversed for the two different photon energies, as evident by the legend.*

We thank the reviewers for pointing this out to us. We have increased the symbol sizes in figure 2 and reduced the number of displayed data points in the bottom panel to make the scatter plot sparser, which makes it easier to distinguish the symbols.

• *Fig. 3: The font size of the legend could be larger, to make the subscripts more easily readable. The fact that the green line shows the asymmetry parameter g should be added into the caption to improve clarity.*

We thank the reviewer for pointing this out. We have adjusted the font size of the legend, and mentioned the green line and right axis in the caption.

We also recommend some minor changes to the text:

• *Line 111: et al. should be italicized.*

We thank the reviewers for pointing this out, and adjusted the formatting.

• *“Note that the α values depend sensitively on the size and refractive index of the aerosol particles at the used radiation wavelength, that directly influence the induced light intensity distribution within the particle.”*

Line 138: it is not clear to me what subject is providing the influence: the wavelength, or the size and refractive index. If it is only the wavelength, then it should be “influences”.

We agree with the reviewers and adjusted the formulation to read:

“Note that the α values depend sensitively on the size and refractive index of the aerosol particles at the used radiation wavelength, because both directly influence the induced light intensity distribution within the particle.”

• *“As a first step we characterize the light intensity distribution within the tyrosine nanoparticles that are necessary to produce the observed shadowing effects; whatever is the polarization of the ionizing radiation.”*

Line 242: the second part of the sentence is confusing and should be rephrased.

We agree that the formulation was not ideal and rephrased:

As a first step we characterize the light intensity distribution within the tyrosine nanoparticles that are necessary to produce the observed shadowing effects, which is independent of the polarization of the ionizing radiation.

• *Finally, centering the equations on page 6 might help with the reading flow.*

We thank the reviewers for pointing out the missing alignment and adjusted accordingly.

Reviewer #3:
